# The effects of queen mandibular pheromone on nurse-aged honey bee (*Apis mellifera*) hypopharyngeal gland size and lipid metabolism

Angela Oreshkova[1]*, Sebastian Scofield[1], Gro V. Amdam[1,2]

1 School of Life Sciences, Arizona State University, Tempe, AZ, United States of America, 2 Faculty of Environmental Sciences and Natural Resource Management, Norwegian University of Life Sciences, Aas, Norway

* aoreshko@asu.edu

## Abstract

Queen honey bees (*Apis mellifera*) release Queen Mandibular Pheromone (QMP) to regulate traits in the caste of female helpers called workers. QMP signals the queen's presence and suppresses worker reproduction. In the absence of reproduction, young workers take care of the queen and her larvae (nurse tasks), while older workers forage. In nurses, QMP increases lipid stores in abdominal fat tissue (fat body) and protein content in hypopharyngeal glands (HPG). HPG are worker-specific head glands that can synthesize proteinaceous jelly used in colony nourishment. Larger HPG signifies ability to secrete proteinaceous jelly, while shrunken glands characterize foragers that do not make jelly. While it is known that QMP increases abdominal lipid stores, the mechanism is unclear: Does QMP make workers consume more pollen which provides lipids, or does QMP increase lipogenic capacity? Here, we measure abdominal lipogenic capacity as fatty acid synthase (FAS) activity while monitoring abdominal protein content and HPG size in caged workers. Cages allow us to rigorously control worker age, pheromone exposure, and diet. In our 2-factorial design, 3- vs. 8-day-old workers (age factor) were exposed to synthetic QMP or not (pheromone factor) while consuming a lipid deficient diet. We found that QMP did not influence abdominal FAS activity or protein content, but QMP still increased HPG size in the absence of dietary lipids. Our data revealed a positive correlation between abdominal protein content and HPG size. Our findings show that QMP is not a strong modulator of lipogenic capacity in caged worker bees. However, our data may reflect that QMP mobilizes abdominal protein for production of jelly, in line with previous findings on effects of honey bee Brood Pheromone. Overall, our study expands the understanding of how QMP can affect honey bee workers. Such insights are important beyond regulatory biology, as QMP is used in various aspects of beekeeping.

## 1. Introduction

Pheromones are chemical signals used for communication between members of the same species [1]. In insects, some of the extensively studied roles of pheromones include attraction of

**Data Availability Statement:** e have deposited our raw data to Figshare and down below we have provided links to each dataset/figure associated with the manuscript. 1. Supplementary Information

Fig 2 (doi: https://doi.org/10.6084/m9.figshare.26280910.v1 ) 2. Supplementary Information Fig 1 (doi: https://doi.org/10.6084/m9.figshare.26263769.v2 ) 3. Supplementary Information File (doi: https://doi.org/10.6084/m9.figshare.26263250.v1 ) 4. Average mortality rate measured in percentage (%) in each cage (doi: https://doi.org/10.6084/m9.figshare.24848235.v1 ) 5. Hypopharyngeal Gland Photos (doi: https://doi.org/10.6084/m9.figshare.24164346.v1 ) 6. R code script for the analysis of the manuscript (doi: https://doi.org/10.6084/m9.figshare.24164343.v2 ) 7. R code history (doi: https://doi.org/10.6084/m9.figshare.24164331.v1 ) 8. Abdominal metrics dataset (abdominal fatty acid synthase values in nmol/min, abdominal protein values in mg, and normalized fatty acid synthase values in nmol/min/mg) (doi: https://doi.org/10.6084/m9.figshare.24164316.v1 ) 9. Average hypopharyngeal acini area measured in mm2 (doi: https://doi.org/10.6084/m9.figshare.24164265.v1 ) 10. Average protein paste and sucrose solution consumption measured in mg/bee/day (doi: https://doi.org/10.6084/m9.figshare.24158433.v1 ).

**Funding:** This project was supported by the Barrett Honors College at Arizona State University. URL: https://barretthonors.asu.edu/. The Honors Thesis and Honors Project Funding grant (TPF-FY23-3170019968) was awarded to AO. The funders did not and will not have a role in study design, data collection and analysis, decision to publish, or preparation of the manuscript.

**Competing interests:** The authors have declared that no competing interests exist.

mates, signaling of reproductive status, maintenance of social hierarchy, and recognition of kin [1]. Insect pheromones are categorized as releaser and/or primer pheromones. Releasers elicit immediate behavioral responses within seconds or minutes, while primers produce long-term endocrine or reproductive changes that may take days to operate in full effect [2, 3]. The long-term physiological effects that primer pheromones exhibit largely influence insect colony organization, caste structure, and the division of labor [4]. One of the most intricate and highly studied insect pheromonal systems is that of honey bees (*A. mellifera*), with approximately 50 substances that have a biologically relevant role in colony life [5].

Each honey bee colony consists of a reproductive queen, non-reproductive females known as workers, and males known as drones [6]. Adult worker honey bees show a division of labor called age polyethism, in which individuals perform different social tasks as they age. Young workers (first 2–3 weeks after emerging from pupation) work inside the nest to groom and nourish all colony members, while older workers venture outside to collect pollen, nectar, water and propolis [7, 8]. This age polyethism is accompanied by physiological specializations, such as the young workers (nurses) having larger abdominal lipid stores and hypopharyngeal glands (HPGs) to support their role in colony nourishment, while foragers are leaner and rely on sugars to fuel their flights [7–9].

Queen mandibular pheromone (QMP), released by the queen bee, influences worker division of labor and physiology [4]. QMP signals the presence of a queen and is categorized as both a releaser and a primer. As a releaser, QMP prompts young worker bees to groom the queen through a retinue response and further spread the pheromone in the colony [3, 5]. As a primer, QMP suppresses genes associated with foraging while activating genes associated with nursing [10]. These are a few examples of the well-known functions of QMP. Past studies show that QMP can increase abdominal lipid stores and the size of HPGs in nurse-aged workers [11–14]. HPGs are paired head glands that can produce proteinaceous jelly [4]. They contain a higher concentration of lipids than other tissues in the head, and bees fed diets high in lipids gain abdominal fat and develop larger HPGs [15, 16]. Furthermore, there has been research that exposure to QMP increases both pollen and sucrose solution consumption in nurse-aged bees [13]. The variable macronutrient composition (lipids and amino acids) of pollen as well as the potential contamination of pesticides makes it difficult to tease apart how QMP acts to influence physiology in these studies [13, 15, 16]. Are the QMP-induced increases in lipid stores and HPG size behaviorally modulated by increasing consumption of lipid-containing pollen, or are they due to effects on metabolic pathways regulating lipid and protein synthesis and storage?

To assess a possible increase in lipid synthesis in the fat body of workers exposed to QMP, we measured the activity of fatty acid synthase (FAS), an enzyme that catalyzes a rate-limiting step of *de novo* lipogenic capacity that combines malonyl-CoA with acetyl-CoA to produce long-chain fatty acids, thus serving as a quantitative metric of lipogenic capacity, the capacity of a tissue to synthesize lipids *de novo* [17]. These long-chain fatty acids are stored as triglycerides in honey bees and most other animals [18]. Thus, in this study we use a FAS activity assay to test whether QMP influences lipogenic capacity in nurse-aged worker bees. Additionally, since nurse bees have high fat body lipids and lipoproteins synthesized there are transported to the HPGs to be used for jelly synthesis [19], we were interested in measuring if there was any relationship between fat body lipogenic capacity and HPG development. Since past studies have shown lipogenesis is more quantitatively important when bees are not fed lipids [20] and that HPG size strongly responds to dietary lipids [16], we fed bees an artificial, lipid-deficient diet, eliminating the possibility that any effect of QMP is due to increased lipid consumption. We hypothesized 1) that QMP increases lipogenic capacity in nurses' abdominal fat bodies (functionally homologous to liver and white adipose tissue), explaining the larger lipid stores

reported previously [11, 12], and 2) that this increased fat body lipogenesis functions to support HPG development so that nurses can fulfill their social role within the colony.

Our specific design was a 2 factorial cage experiment with an age factor and a pheromone factor: 3-day-old (representing developing bees not yet physiologically competent to become nurses or foragers) or 8-day-old (representing bees at peak nursing age) were exposed to synthetic QMP, or not. We chose to measure these two age groups since factors that affect nurse-bee physiology like QMP sometimes have different effects on 3 and 8-day old bees [11] and measuring both timepoints thus gives a better chance of identifying a possible effect. FAS activity in the fat body was normalized by measuring the amount of abdominal protein per nurse-aged bee. We monitored HPG size by measuring the area of the glands' acini. For each cage, we also monitored the depletion of the lipid-deficient diet as a function of worker mass, to control for whether QMP influenced food consumption.

## 2. Materials and methods

### 2.1 Honey bees

The experiments were performed in September through November of 2022 at Arizona State University Campus, Tempe. To start, three frames from 3 different hives were collected, sealed in a mesh cage, placed in an incubator at 33°C, and kept humidified using an open dish of water at the bottom of the incubator. After 24 h, newly emerged bees were collected and placed in small (16 × 12 × 9 cm) Plexiglass and mesh cages (30 bees/cage). Bees were fed 30% w/v sucrose solution *ad libitum* in 20 ml syringes with their tips cut off and placed at the top of the cage as described previously [21]. To feed the bees sufficient protein for development of HPGs without feeding them lipids, we also fed the bees artificial diets using soy as a protein source as described previously [22]. We chose to use artificial diets because many studies have shown strong effects of dietary pollen on worker physiology [11], but pollen composition is highly variable and difficult to standardize [23] and pollen may be contaminated with pesticides [24]. To eliminate these issues and provide greater insight into specific nutrients in pollen, a number of recent studies have used artificial diets [16, 25, 26]. A paste containing 20% total protein was made with 21.47% soy protein isolate (MP Biomedical) and 78.26% honey and fed to the bees using 1.5 mL Eppendorf feeders placed at the bottom of each cage (Fig 1). Consumption of sucrose solution and protein paste was calculated by weighing the feeders every 24 hours. Cage experiments using cohorts of approximately 30 bees per cage maintained in an incubator are standard practice when investigating the role of pheromones and diet on physiological parameters and have been used in a number of different studies [12, 27, 28].

A 2 × 2 factorial experimental design was used, and each cage was assigned to an age-group (3 and 8-day-old bees) and a treatment group of QMP+ or QMP−. Each combination of age and presence/absence of QMP was assigned to one cage and this was replicated 3 times, yielding a total of 12 cages (1 cage × 4 treatment groups × 3 replicates; Fig 2). This level of replication has been used for other cage experiments (for example [29]) and is sufficient for a study in which the focus is not on cage level metrics but on sampling bees within each cage. To produce bees of specific ages, cages were set up either 3 or 8 days prior to each sampling day. The cages were maintained at 33°C in a dark, humidified incubator and mortality was recorded every day in each cage. Cages also received the QMP treatment for the duration of those 3 or 8 days. Synthetic QMP ("TempQueen"; Betterbee Inc.) was presented as a slow-release strip placed at the bottom of the QMP+ cages, while QMP− cages did not receive a QMP strip, as in an earlier study [30]. This dose and application of QMP is standard practice for cage experiments examining the role of QMP [11, 31–33]. The strip was used according to the manufacture instructions and is equal to 10 queen equivalents of the 5-component blend of QMP. It

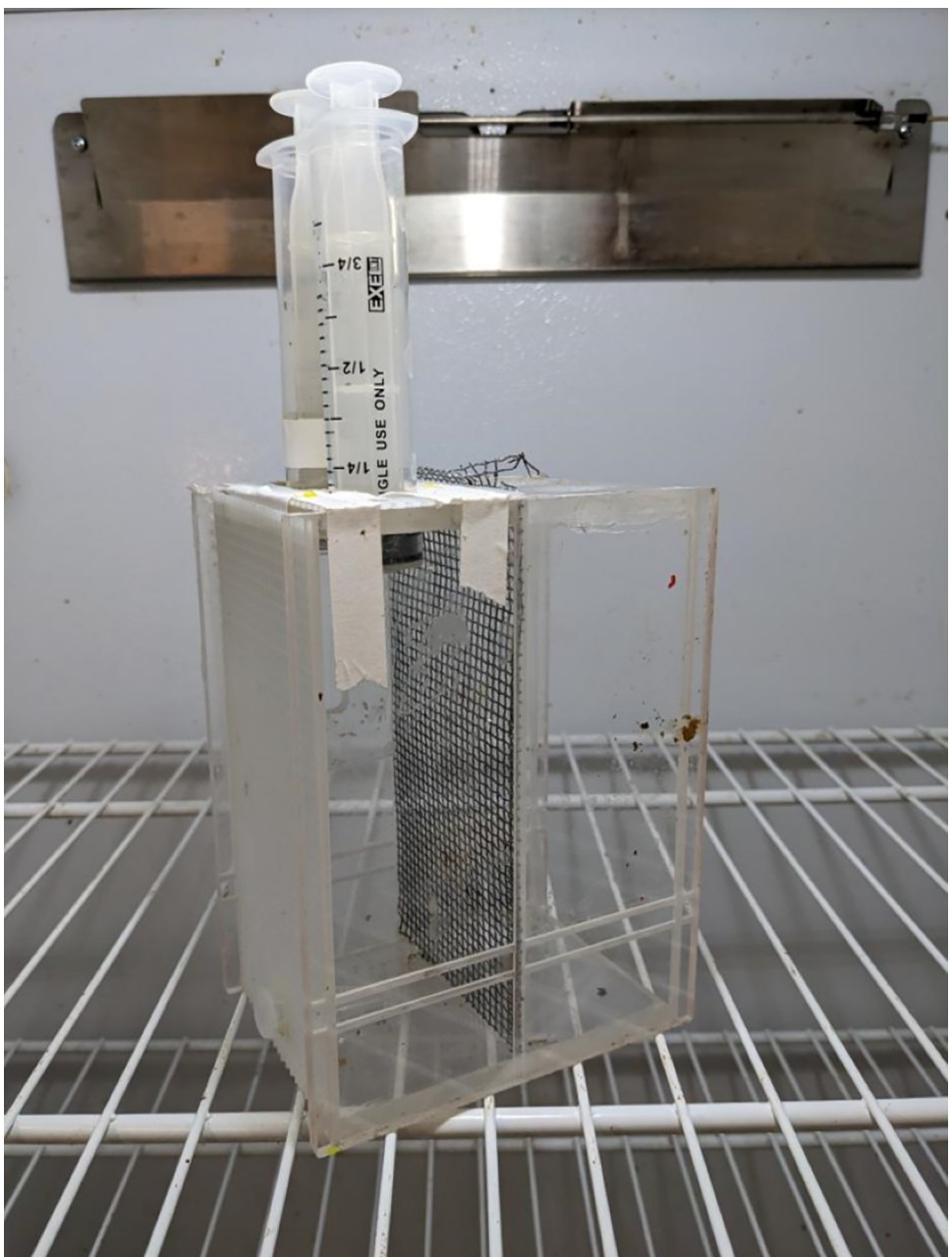

**Fig 1. Cage design used for the queen mandibular pheromone treatments.** The dimensions measured were 16 cm × 12 cm × 9 cm. Because approximately half the cage was used, the effective depth was 6 cm.

mimics exposure to natural QMP, a mixture of 9-keto-2-(E)-decenoic acid (ODA), the enantiomers of 9-hydroxy-2-(E)-decenoic acid (9-HDA)(88% R-(−) and 12% S-(+)), methyl p-hydroxybenzoate (HOB), and 4-hydroxy-3-methoxyphenylethanol (HVA) [2]. One queen equivalent represents the amount of QMP a mated queen will produce in a 24-hour period and contains 200 mg of ODA, 80 mg of 9-HDA, 20 mg of HOB, and 2 mg of HVA [34]. QMP is not volatile as it is spread by honey bee workers via trophallaxis, antennation, and cuticular contact [30]. Thus, all cages were kept in the same incubator. During sampling, bees were anesthetized on ice, euthanized, and dissected for subsequent analyses.

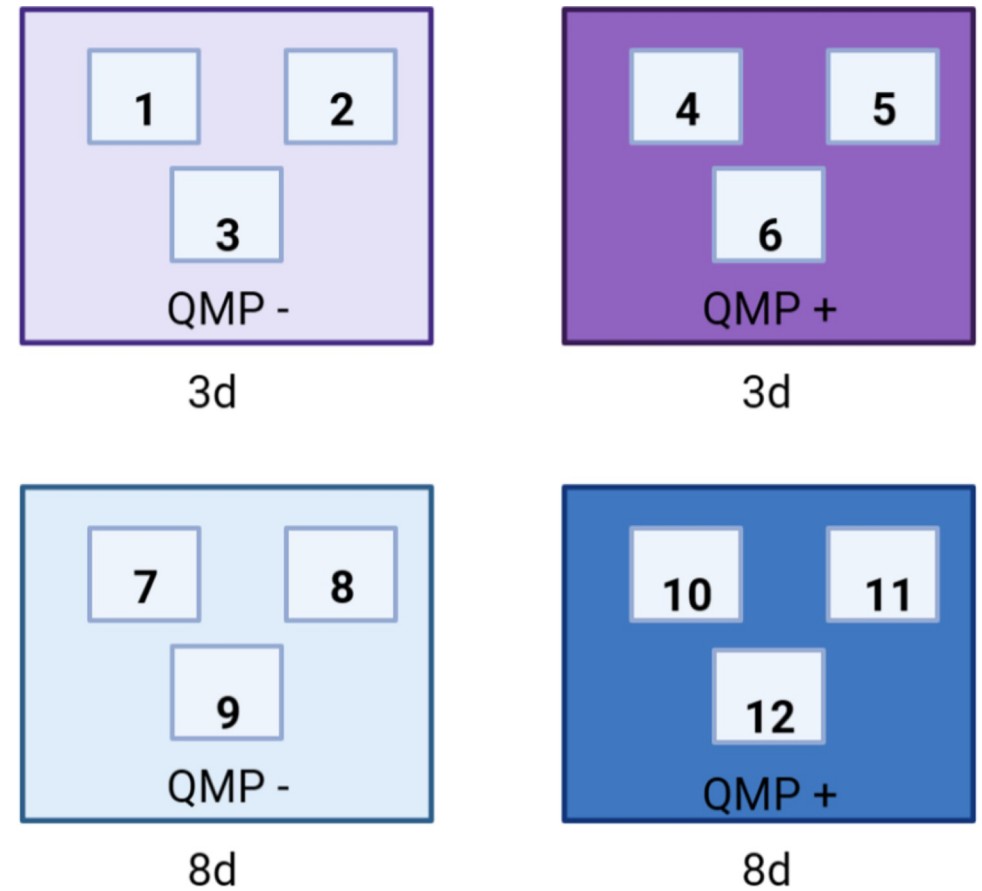

**Fig 2. Pictorial depiction of the 2-factorial design used in the study.** The 2-factorial design that represented all combinations of the presence vs absence of synthetic queen mandibular pheromone ('QMP') and 3 vs 8-day-old nurse-aged honey bees ('age'). This resulted in 4 experimental treatments, each applied to one cage of bees. The design was replicated three times, resulting in three replicates. The small boxes labelled inside each large box represent the cage identity (12 total cages used).

## 2.2 Fatty acid synthase (FAS) activity measurements

Fat body fatty acid synthase (FAS) activity was measured using a previously described method with minor modifications [35]. Briefly, the abdominal carcass (complete abdominal cuticle with adhering fat body tissue minus the stinger, ovaries, gut, and crop) was isolated from workers, pooled in pairs of two, homogenized in phosphate-buffered saline containing protease inhibitors (11697498001; Roche Applied Science; Indianapolis; IN; USA), sonicated for 30 seconds, and centrifuged at $10,000 \times g$ for 5 minutes at 4°C. The resulting supernatant was collected and assayed immediately in a 96-well microplate. In each well, 33.3 μL of supernatant was mixed with 163.3 μL of 2.0 M potassium phosphate buffer, pH 7.1, 16.7 μl of 20 mM dithiothreitol, 20 μl of 0.25 mM acetyl-CoA, 16.7 μl of 60 mM EDTA. To initiate the reaction, 33.3 μl of 0.39 mM malonyl-CoA was added to each well. FAS activity was measured as the oxidation of NADPH at 340 nm and 37°C using a UV/VIS spectrophotometer (Synergy H1 Multimode Reader, BioTek). A background correction was made for the oxidation of NADPH in the absence of malonyl-CoA. Background and sample wells were both measured in duplicate. This assay was replicated 3 times. In each replicate, 2 bees were pooled for each biological sample and 6 samples totaling 12 bees were collected from each cage for all four treatment conditions. This totaled 144 abdomens sampled across three replicates. FAS activity was calculated

as nmol of NADPH oxidized/min/abdomen. Additionally, FAS activity was normalized by the amount of total soluble protein in each sample, measured with a BCA Assay Kit (Thermo Scientific) according to manufacturer's instructions, giving FAS activity in nmol/min/mg. For a more complete picture of the metabolic state of the bee, we report total abdominal FAS activity as well as FAS activity normalized to mg of abdominal protein. Because our dissection approach collects the entire abdominal cuticle and attached fat body, it produces a metric of total lipogenic capacity per abdomen, which is similar to past studies that have measured total lipids per abdomen using the same dissection technique [7, 12, 15, 16]. To control for potential differences in protein extraction efficiency, we also measure protein concentration in each sample and normalize FAS activity relative to extracted protein. Comparing results from the two metrics makes it clearer when treatment group differences are due to changes in the quantity of activated FAS or due to changes in the normalization factor itself.

## 2.3 Hypopharyngeal gland measurements

To determine whether QMP affects HPG acini size, honey bee heads from the same bees used for FAS activity were flash frozen in liquid nitrogen and kept in a -80˚C freezer until they were dissected. Per treatment group, 18 heads were dissected, resulting in a total of 72 heads. For each head, the HPGs were first dissected into a glass plate with concave deep wells containing 40 µl of 10x Giemsa for 7 min. They were then transferred into a flat microscope slide that contained 60 µl of 1x PBS buffer (37 mM NaCl, 2.7 mM KCl, and 10 mM PO4, pH 7.4) and visualized at 60 to 80x magnification, as previously described [36]. The glands were visualized under a Leica M205C stereoscope with a Leica DFC450 camera using the Leica Applications Suite v4.5 software. A blind observer was then told to select 10 acini per bee with the criteria that the acini were in focus, had clear attachment points to the collecting duct, and were average relative to all the acini in the photo. The area (mm) of those 10 selected acini per bee was measured using ImageJ by the researcher. The areas were then averaged per bee and analyzed as a pooled sample in which the heads that were pooled together corresponded to the abdomens that were pooled together during the FAS analysis.

## 2.4 Statistical analysis

The effects of age and QMP on abdominal FAS activity, abdominal protein, and HPG acini area were preferentially processed with an ANOVA test using three variables: age, treatment, and replicate, and the interaction effects between all three variables. Datasets that were analyzed with ANOVA adhered to its assumptions of normality, estimated by normal probability plots of the datasets and a Shapiro-Wilks test, and homogeneity of variances, determined by a Levene's test. The minimum p value for a significant dataset was 0.05. When assumptions were not met, in the case of normalized FAS activity dataset and protein/sucrose consumption, a non-parametric test was used, specifically, a Kruskal-Wallis test to look at differences between the four treatment groups. The non-parametric test was followed by a Dunn's Post hoc test if needed. Replicates were included in the ANOVA to control for differences between replicates. A total of three replicates were performed for the study and each replicate reflects a combination of cage/day and FAS assay plate variation. For each parameter analyzed with either a parametric or non-parametric test, replicate effects were included in an appropriate model. When using ANOVA, replicate was added as a factor including interaction effects between replicate and other factors. For the normalized FAS data that did not meet the assumptions for ANOVA, a Kruskal-Wallis test was used to determine if there were any significant differences between replicates. Since the pattern was similar across the three replicates (S2C Fig), the data were pooled to look for differences between treatment groups. The relationships between HPG acini size and abdominal protein and abdominal FAS activity were analyzed using linear regression. Because the data deviated significantly from the assumptions

of normality, the relationship between abdominal protein and FAS activity was analyzed using a Kendall-Theil Sen Siegel non-parametric linear regression. All statistics were analyzed with R version 4.3.1, Rstudio, and the packages 'FSA', 'performance', 'mblm', and 'rcompanion' [37–42]. For all measurements except for consumption, the experimental unit was bees or pooled bees. For consumption metrics, the cage was the experimental unit. A balanced study design was employed for all factors. The sample size for cage level consumption metrics was low at $n = 3$ cages per treatment as this was not the focus of our study and the metrics describe the physiological parameters of the bees. For measuring abdominal FAS activity, normalized FAS activity, abdominal protein, and HPG acini area, 2 bees were pooled for each biological sample and 6 samples totaling 12 bees were collected from each cage for all four treatment conditions. This totaled 72 biological samples pooled from 144 abdomens as three replicates were performed. Thus, the sample size was $n = 18$ pooled samples per treatment group for these three parameters. The sample size for the regression analyses was $n = 35$ pooled samples as there was a dissection issue for one of the bees and only half of the HPGs of the 24 pooled samples per replicate were measured, except for the case of abdominal protein and abdominal FAS which was $n = 72$ (144 total abdomens).

## 3. Results

### 3.1 Food consumption and mortality

Protein paste consumption did not significantly differ between any of the four treatment groups (Kruskal-Wallis, chi-squared = 2.6923, df = 3, $P = 0.4415$, $n = 3$ cages; Fig 3).

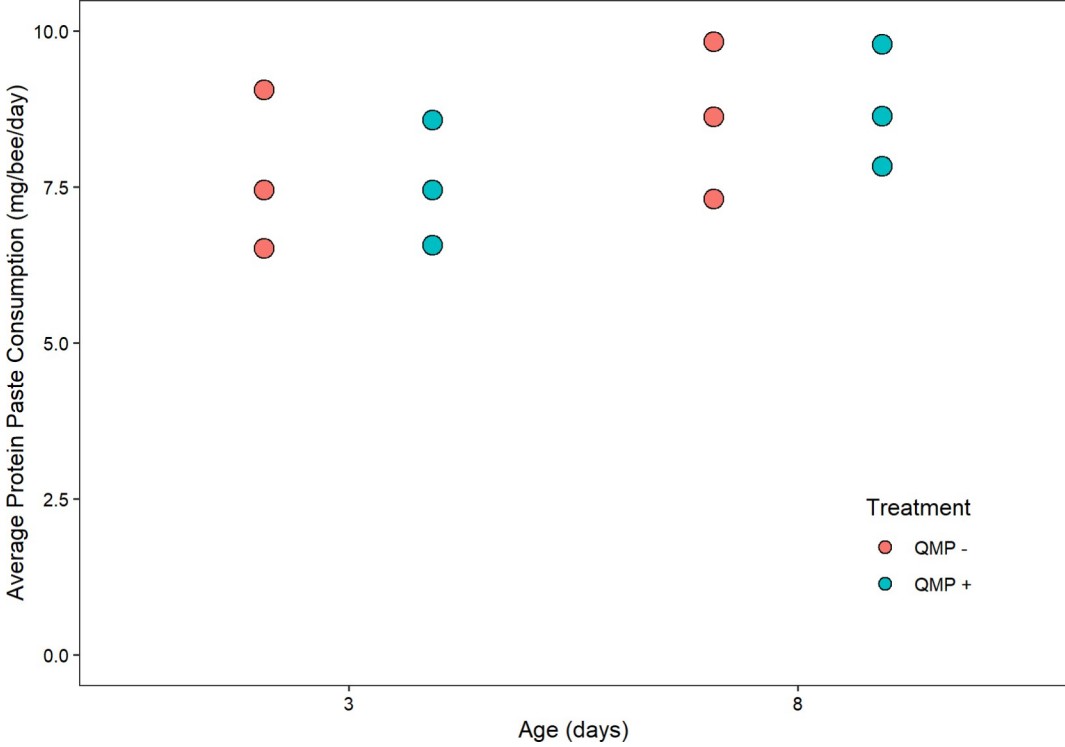

**Fig 3. The relationship between age(days)/QMP and protein consumption in mg/bee/day.** A non-parametric Kruskal-Wallis test showed that there were no statistical differences between each treatment's protein consumption ($P = 0.4415$). Each point represents the average protein paste consumption for one cage ($n = 3$ cages). The boxplot shows the mean (middle black line of box), the interquartile range (box boundaries), and the minimum and maximum values of the distribution. Outliers are points outside the maximum and minimum of the distribution.

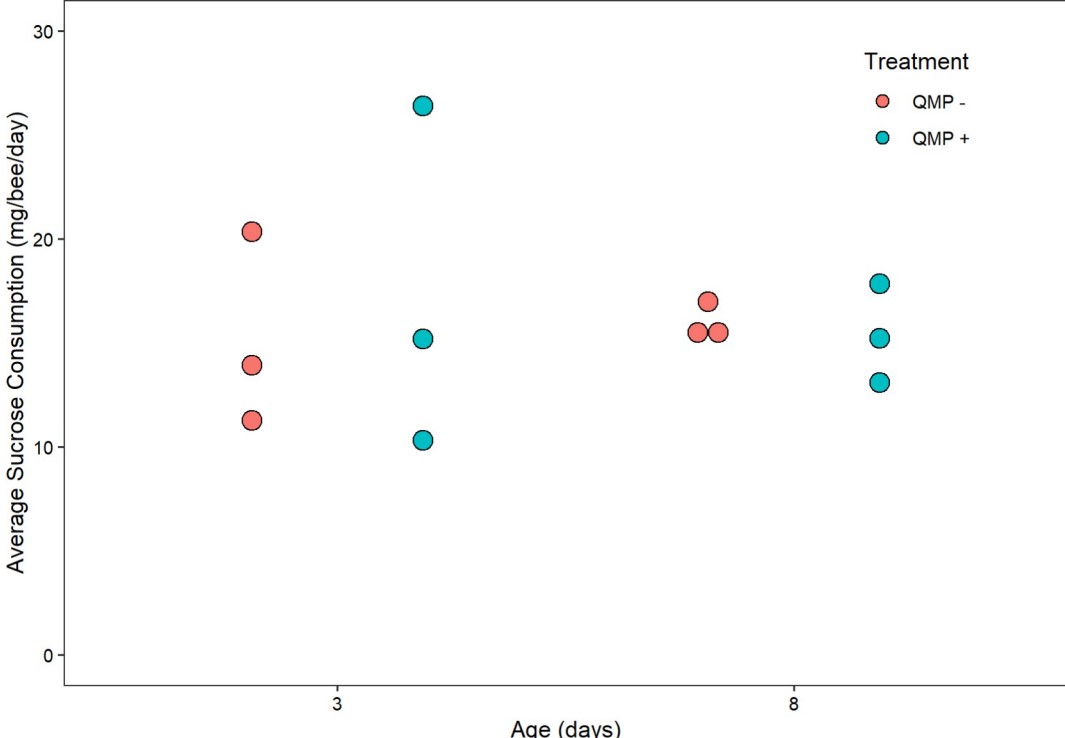

**Fig 4. The relationship between age(days)/QMP and sucrose consumption in mg/bee/day.** A non-parametric Kruskal-Wallis test showed that there were no statistical differences between each treatment's sucrose solution consumption ($P = 0.8894$). Each point represents the average sucrose consumption for one cage ($n = 3$ cages). The boxplot shows the mean (middle black line of box), the interquartile range (box boundaries), and the minimum and maximum values of the distribution. Outliers are points outside the maximum and minimum of the distribution.

Furthermore, sucrose consumption did not significantly differ between any of the four treatment groups (Kruskal-Wallis, chi-squared = 0.63041, df = 3, $P = 0.8894$, $n = 3$ cages; Fig 4). Protein consumption did not differ significantly between replicates (chi-squared = 3.7308, df = 2, $P = 0.1548$) but sucrose consumption did differ significantly between replicates (chi-squared = 7.4974, df = 2, $P = 0.02355$). Mortality rates were low (less than 7%) and within accepted ranges for cage experiments [21] for all ages and did not differ by QMP treatment (Kolmgorov-Smirnov, $D(10) = 0.167$, $P = 1$; S1 Fig).

## 3.2 Abdominal FAS activity and abdominal protein

Age ($F_{1,60} = 2.010$, $P = 0.16140$) and QMP treatment ($F_{1,60} = 0.698$, $P = 0.40668$) had no significant effect on abdominal FAS activity in nmol NADPH oxidized per minute per abdomen (Fig 5). Furthermore, there was no interaction effect between age × QMP treatment ($F_{1,60} = 0.484$, $P = 0.48931$), QMP treatment × replicate ($F_{2,60} = 0.962$, $P = 0.29039$), or age × QMP treatment × replicate ($F_{2,60} = 0.726$, $P = 0.48796$). However, there were significant differences in abdominal FAS activity between the three replicates ($F_{2,60} = 29.323$, $P < 0.001$; S2A Fig) and a significant interaction effect between age × replicate ($F_{2,60} = 5.152$, $P = 0.00862$). To normalize the FAS activity assay, the abdominal protein (in mg) of each pooled sample was measured. Abdominal protein was significantly higher in 8-day-old than 3-day-old bees ($F_{1,60} = 35.693$, $P < 0.001$; Fig 6) and significantly different between replicates ($F_{2,60} = 36.000$ $P < 0.001$; S2B Fig); it was not significantly affected by QMP treatment ($F_{1,60} = 0.032$, $P = 0.8596$), age × QMP treatment ($F_{1,60} = 1.057$, $P = 0.3081$), age × replicate ($F_{2,60} = 2.809$, $P = 0.0682$),

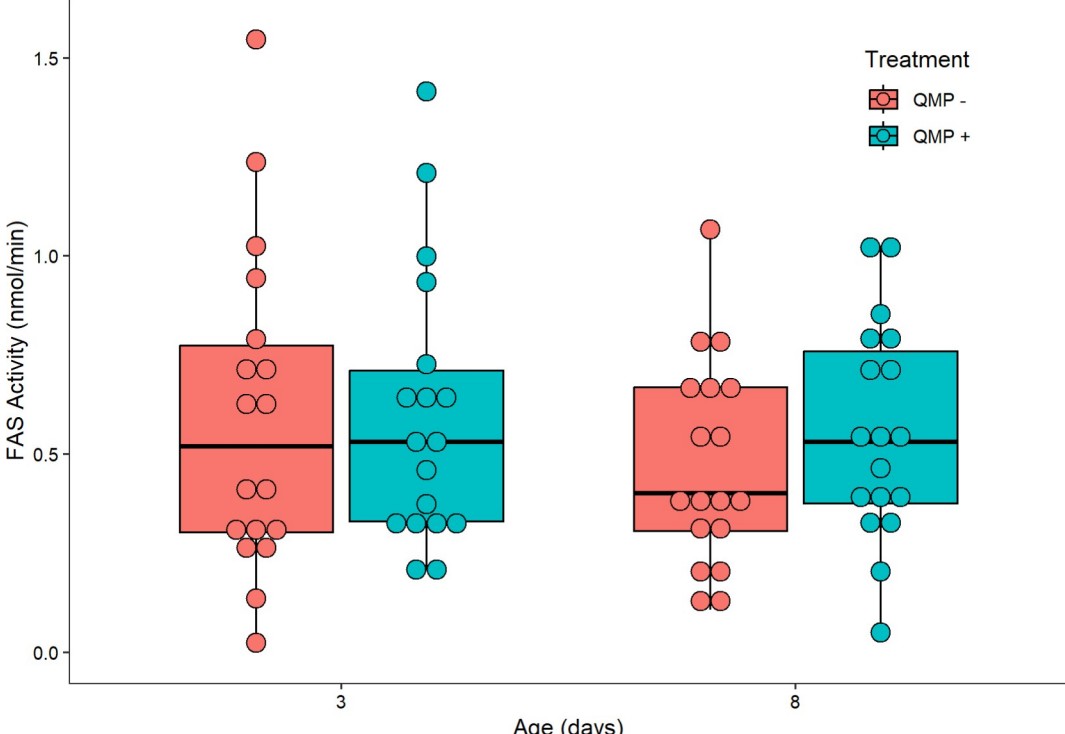

**Fig 5. The relationship between age/QMP treatment and abdominal FAS activity (NADPH oxidized) in nmol/min.** Age ($F_{1,60}$ = 2.010, $P$ = 0.16140) and QMP treatment ($F_{1,60}$ = 0.698, $P$ = 0.40668) had no significant effect on abdominal FAS activity in nmol NADPH oxidized per minute per abdomen. Furthermore, there was no interaction effect between age × QMP treatment ($F_{1,60}$ = 0.484, $P$ = 0.48931), QMP treatment × replicate ($F_{2,60}$ = 0.962, $P$ = 0.29039), or age × QMP treatment × replicate ($F_{2,60}$ = 0.726, $P$ = 0.48796). Each point represents the average abdominal FAS activity for each pooled sample of two bees. As there were 6 pools per cage (12 abdomens/bees sampled) and three replicates performed, the sample size is $n$ = 18 pooled samples. The boxplot shows the mean (middle line of box), the interquartile range (box boundaries), and the expected variation of the data (whiskers, calculated as 1.5 times the interquartile range past the top and bottom of the box). Outliers are points outside the expected variation in the data.

treatment × replicate ($F_{2,60}$ = 0.717, $P$ = 0.4925), or age × treatment × replicate ($F_{2,60}$ = 1.240, $P$ = 0.2968).

### 3.3 FAS activity normalized to abdominal protein

Using the abdominal protein quantity, FAS activity was normalized per pooled sample in nmol of NADPH oxidized per min per mg of protein. Normalized FAS activity significantly differed between the three replicates (chi-squared = 23.87, df = 2, $P$ < 0.001; S2C Fig) and the four treatment groups (Kruskal-Wallis, chi-squared = 9.3498, df = 3, $P$ = 0.02498; Fig 7). Comparisons between the four treatment groups showed that only the 3-day-old QMP + and 8-day-old QMP–groups differed significantly from each other (Dunn's *post-hoc* test, $P_{adj}$ = 0.0397).

### 3.4 Average HPG acini area

Mean HPG acini area was significantly higher in 8-day-old than 3-day-old bees ($F_{1,60}$ = 27.780, $P$ < 0.001; Fig 8) and in bees treated with QMP ($F_{1,60}$ = 29.156, $P$ < 0.001) and was significantly different between replicates ($F_{2,60}$ = 6.853, $P$ = 0.00209; S2D Fig). The interaction effect age × treatment × replicate was significant ($F_{2,60}$ = 4.457, $P$ = 0.01568); the other interaction effects of age × QMP treatment ($F_{1,60}$ = 0.568, $P$ = 0.45392), age × replicate ($F_{2,60}$ = 1.338,

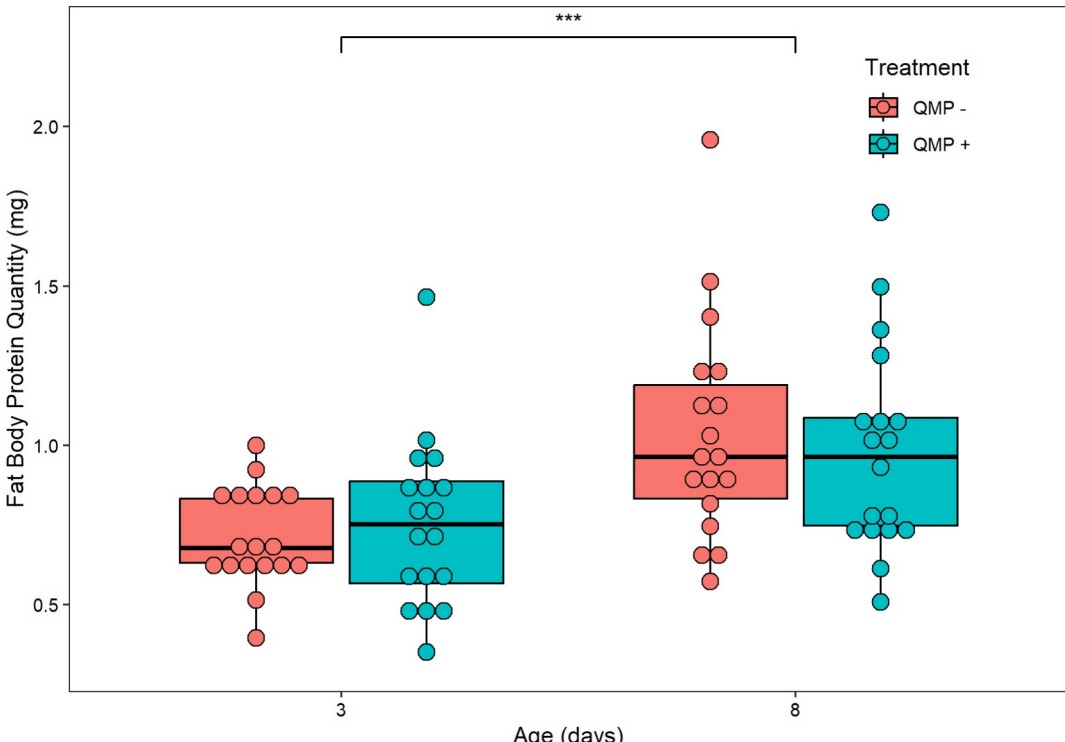

**Fig 6. The effect of age/QMP treatment and abdominal protein quantity in mg.** Abdominal protein was significantly higher in 8-day-old than 3-day-old bees ($F_{1,60}$ = 35.693, $P$ < 0.001). It was not significantly affected by QMP treatment ($F_{1,60}$ = 0.032, $P$ = 0.8596), age × QMP treatment ($F_{1,60}$ = 1.057, $P$ = 0.3081), age × replicate ($F_{2,60}$ = 2.809, $P$ = 0.0682), treatment × replicate ($F_{2,60}$ = 0.717, $P$ = 0.4925), or age × treatment × replicate ($F_{2,60}$ = 1.240, $P$ = 0.2968). Each point represents the average abdominal protein for each pooled sample of two bees. As there were 6 pools per cage (12 abdomens/ bees sampled) and three replicates performed, the sample size is $n$ = 18 pooled samples. The boxplot shows the mean (middle line of box), the interquartile range (box boundaries), and the expected variation of the data (whiskers, calculated as 1.5 times the interquartile range past the top and bottom of the box). Outliers are points outside the expected variation in the data. The asterisk indicates significant differences (*** denotes $P$ < 0.001).

$P$ = 0.27004), and treatment × replicate ($F_{2,60}$ = 2.369, $P$ = 0.10226) did not significantly affect mean HPG acini area.

## 3.5 Relationship between HPG size and abdominal metrics

Mean HPG acini size was significantly positively predicted by abdominal protein quantity ($F_{1,33}$ = 12.66, adjusted $R^2$ = 0.2382, $P$ = 0.00116; Fig 9) and by abdominal FAS activity ($F_{1,33}$ = 5.819, adjusted $R^2$ = 0.1181, $P$ = 0.02123; Fig 10) but not by normalized FAS activity ($F_{1,33}$ = 0.944, adjusted $R^2$ = -0.001558, $P$ = 0.3379). Abdominal FAS activity significantly increased with an increase in abdominal protein quantity (Kendall-Theil Sen Siegel linear regression: estimate = 1.5049 ± 1.4016, $V$ = 2265, Efron's pseudo $R^2$ = 0.0498, $P$ < 0.001; Fig 11). Refer to Table 1 for the regression analysis of $P$ values between all combinations of abdominal protein, abdominal FAS activity and HPG acini area.

## 4. Discussion

In this study, we explored if QMP can increase fat body lipids and HPG size in nurse-aged worker bees fed a lipid-deficient diet. We show that workers exposed to synthetic QMP have larger HPGs on average, but we did not detect a change in the bees' lipogenic capacity. In

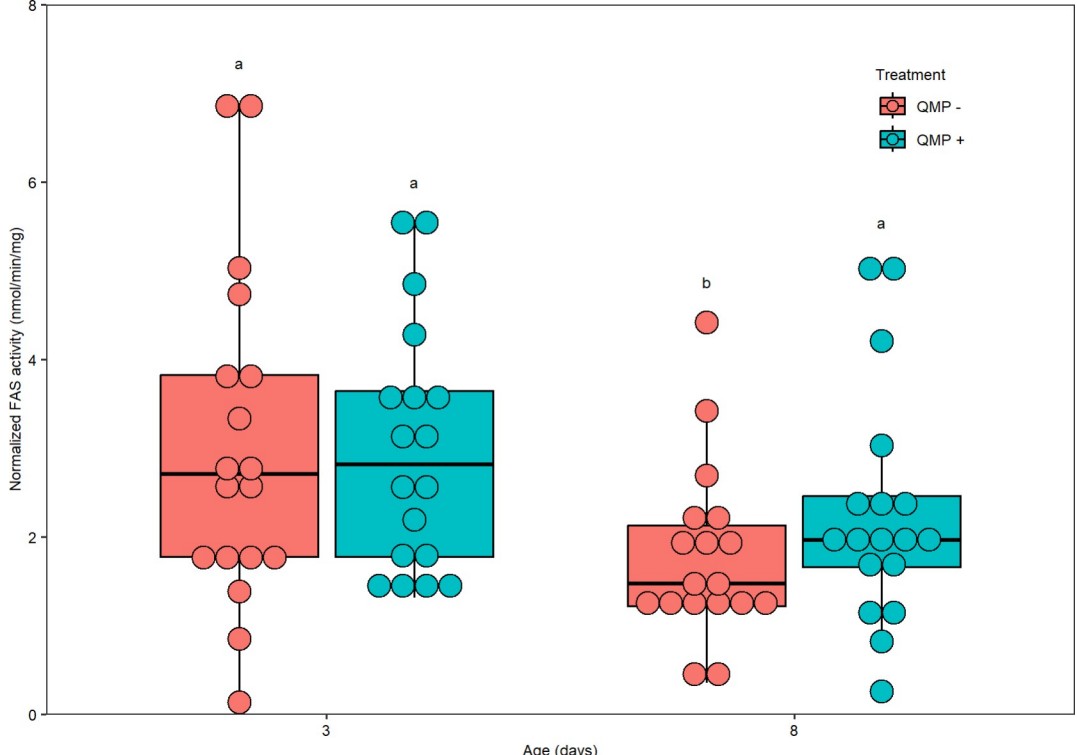

**Fig 7. The effect of age/QMP treatment and normalized FAS activity in nmol/min/mg.** Normalized FAS activity significantly differed between the four treatment groups (Kruskal-Wallis, chi-squared = 9.3498, df = 3, $P$ = 0.02498). Comparisons between the four treatment groups showed that only the 3-day-old QMP + and 8-day-old QMP–groups differed significantly from each other (Dunn's *post-hoc* test, *Padj* = 0.0397). No other treatment groups were significantly different from one another. Each point represents the average normalized FAS activity for each pooled sample of two bees. As there were 6 pools per cage (12 abdomens/bees sampled) and three replicates performed, the sample size is $n$ = 18 pooled samples. The boxplot shows the mean (middle line of box), the interquartile range (box boundaries), and the expected variation of the data (whiskers, calculated as 1.5 times the interquartile range past the top and bottom of the box). Outliers are points outside the expected variation in the data. Significantly different treatment groups are represented by different letters.

interpreting our results, it is important to note that there were several significant differences between experimental replicates. In the case of the sucrose consumption data, this is likely due to our small sample size. There were also significant replicate effects within the abdominal FAS activity data, abdominal protein data, and HPG acini size. Since we sourced bees in the experiment from three different colonies, these replicate effects may have been a result of biological variation among bees with different genetic backgrounds, but we cannot rule out contributions of technical variation as well.

During the experiment, we monitored food consumption to determine whether QMP increases HPG size through affecting worker lipogenic capacity or through simply increasing the workers' food consumption. Overall, QMP did not influence the depletion of sucrose solution or protein paste in our cages. This finding is consistent with a previous study in which the sucrose and pollen consumption of nurse-aged bees did not differ between queenright-like treatments (QMP) and queenless-like treatments (no QMP) [14]. However, other work has shown that QMP exposure can increase food consumption in nurse-aged bees fed rich diets that contained pollen, as well as in nurse-aged bees fed with poor diets consisting of just sucrose solution [13]. Our results may have differed from these latter results because we used synthetic QMP strips while Ament and colleagues [13] used 0.1 equivalents of QMP dissolved

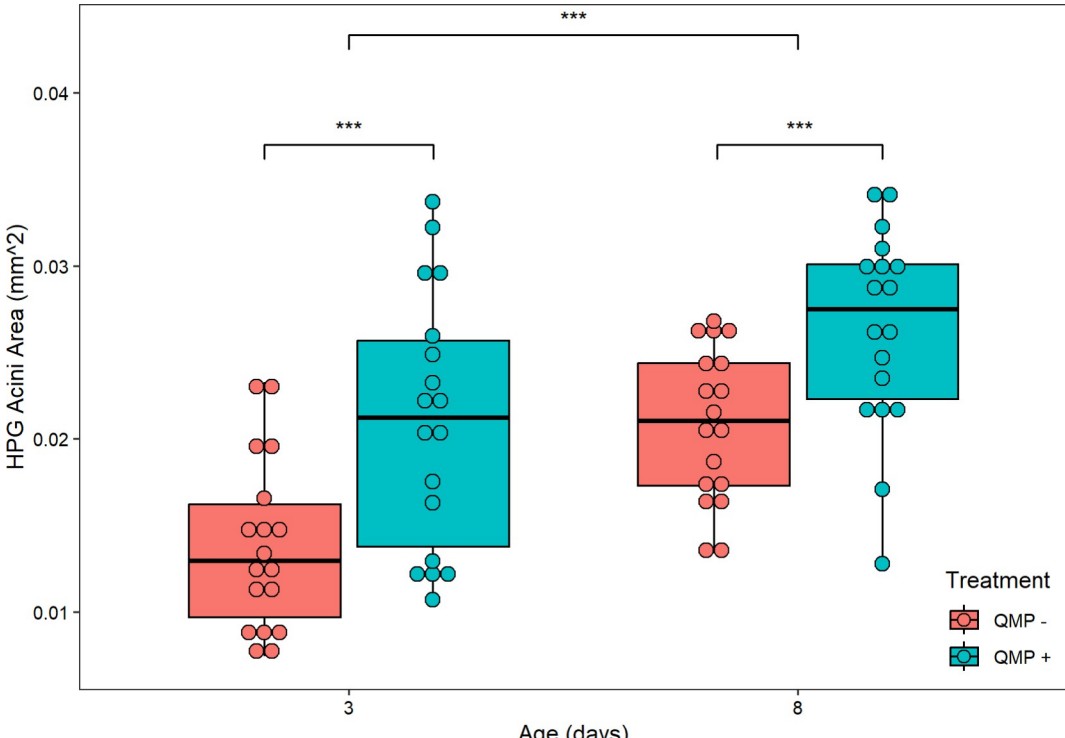

**Fig 8. The effect of age/QMP treatment and average HPG acini area in mm$^2$.** Mean HPG acini area was significantly higher in 8-day-old than 3-day-old bees ($F_{1,60}$ = 27.780, $P$ < 0.001) and in bees treated with QMP ($F_{1,60}$ = 29.156, $P$ < 0.001). The interaction effect age × treatment × replicate was significant ($F_{2,60}$ = 4.457, $P$ = 0.01568); the other interaction effects of age × QMP treatment ($F_{1,60}$ = 0.568, $P$ = 0.45392), age × replicate ($F_{2,60}$ = 1.338, $P$ = 0.27004), and treatment × replicate ($F_{2,60}$ = 2.369, $P$ = 0.10226) did not significantly affect mean HPG acini area. Each point represents the average HPG acini area for each pooled sample of two bees. As there were 6 pools per cage (12 abdomens/bees sampled) and three replicates performed, the sample size is $n$ = 18 pooled samples. The boxplot shows the mean (middle line of box), the interquartile range (box boundaries), and the expected variation of the data (whiskers, calculated as 1.5 times the interquartile range past the top and bottom of the box). Outliers are points outside the expected variation in the data. The asterisks indicate significant differences (*** denotes P < 0.001).

with isopropanol and water on a microscope over slip. There also may be mixed results on whether QMP affects food consumption due to the varying sample sizes and varying diet compositions in all three studies. While our study had a sample size of 3 cages for consumption, Ament and colleagues [13] had a sample size ranging from 6 to 8 cages and Peters and colleagues [14] had a sample size ranging from around 150 to 300 bees, depending on the treatment group. The lack of significant differences in food consumption between our treatments could be due to our small sample size. Thus, our data should be interpreted cautiously. Furthermore, our study used a lipid deficient diet while Ament and colleagues [13] and Peters and colleagues [14] used a combination of diets consisting of only pollen, pollen and sucrose, or only sucrose. A recent study in our lab found that dietary protein but not fat increases FAS activity in 8-day-old bees [43]. There may be some interacting effects between QMP and nutrition which we did not test for.

Age and QMP treatment did not significantly affect FAS activity analyzed per bee, and only age significantly affected FAS activity analyzed per mg of extracted protein. This suggests that QMP does not significantly affect the *de novo* synthesis of lipids in nurse-aged honey bee fat bodies, but caged bees can increase their abdominal protein content as they age. Our larger sample size for mean HPG acini area ($n$ = 18) makes these results more robust than our

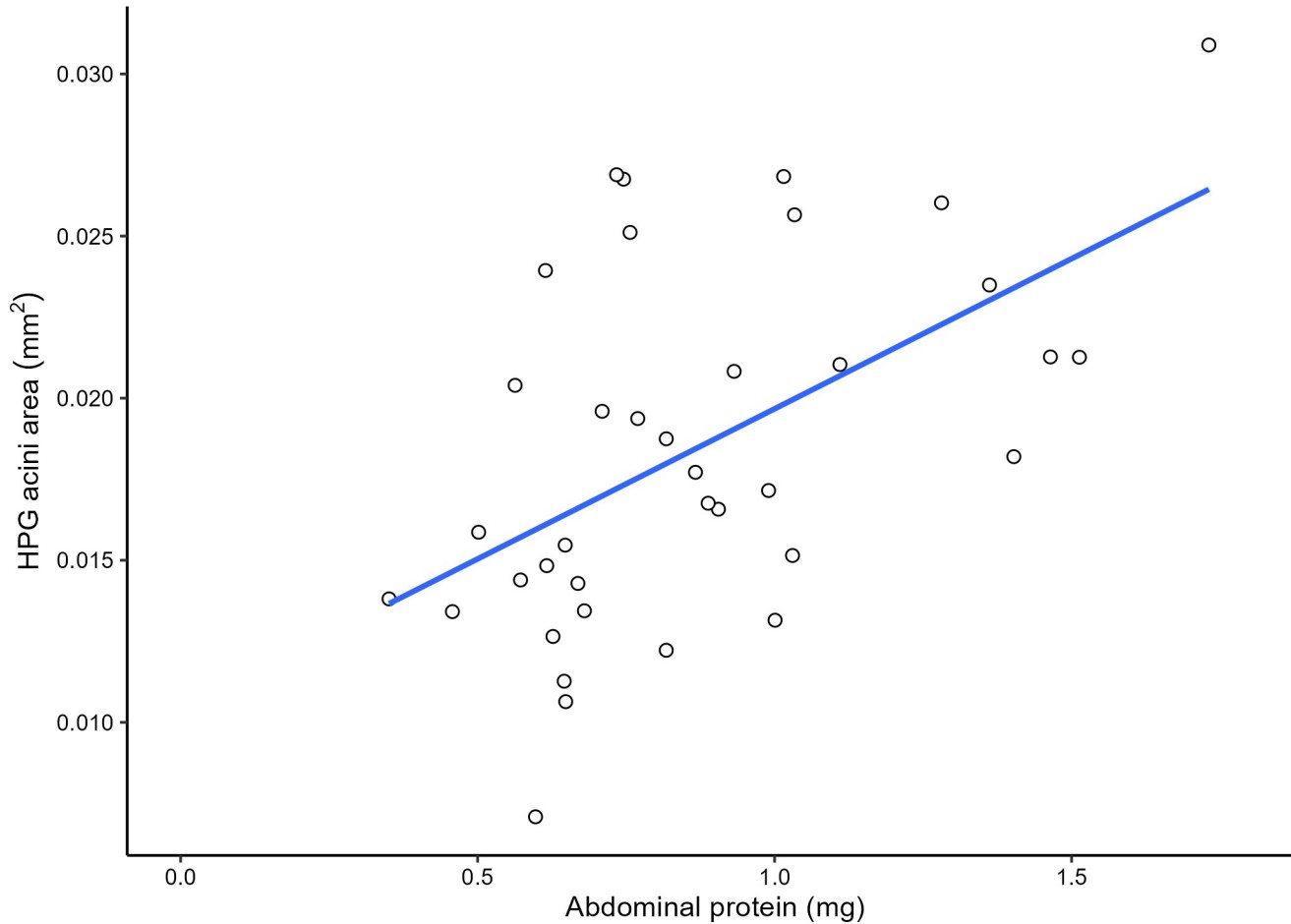

**Fig 9. Regression scatter plot between abdominal protein (mg) and HPG acini area (mm$^2$).** Mean HPG acini size was significantly predicted by abdominal protein quantity ($F_{1,33} = 12.66$, adjusted $R^2 = 0.2382$, $P = 0.00116$). Each point represents the correlation between the pooled sample's (2 bees per) abdominal protein and HPG acini area ($n = 35$ pooled bees). Per treatment, 18 bees were used except for the case of the 3-day-old QMP + treatment group in the first replicate as there was a dissection issue. An additional sample was measured but not included in the regression analysis.

consumption data. In contrast, recent data from our group show that FAS activity does not differ significantly between 3d and 8d bees collected from natural colonies [43]. Our results may have differed due to the environment of both treatments. While the past study experimented on bees within natural hives who were receiving optimal nutrition from being fed by other nurse bees, our study experimented on bees within cages that received less nutrition. Contrary to our findings, it was previously reported that QMP can increase abdominal lipid stores in young bees (3, 4, and 5-day old bees) provided pollen-containing diets, as well as diets consisting only of sucrose solution [11–13]. How can this diversity of results be explained? One possibility is that QMP reduces the activity levels of young bees as shown before [44]. The mechanism underlying the larger lipid stores could be reduced energy expenditure by decreasing the catabolism rate of lipids in young bees. Future work could look at the lipid breakdown rate in bees exposed or not exposed to QMP. If the activity levels and lipid catabolism rates are different between our bees and the bees used in the other studies [11–13] due to varying ages, environment, and diet, this could explain the inconsistency of whether QMP affects abdominal lipid stores.

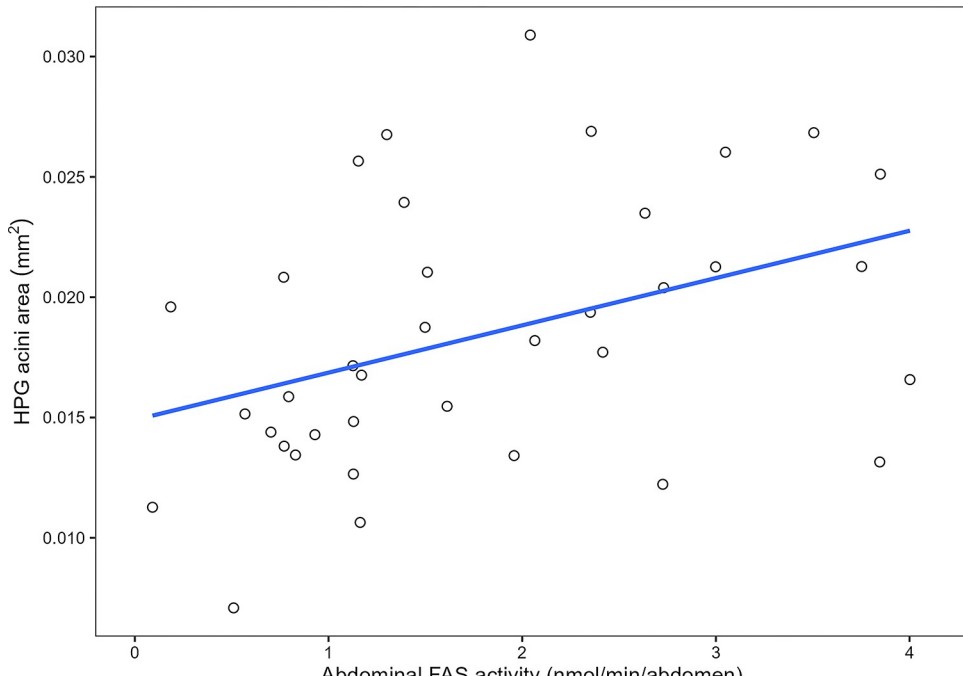

**Fig 10. Regression scatter plot between abdominal FAS (nmol NADPH oxidized/min) and HPG acini area (mm$^2$).**
Mean HPG acini size was significantly predicted by abdominal FAS activity ($F_{1,33}$ = 5.819, adjusted R$^2$ = 0.1181, $P$ = 0.02123). Each point represents the correlation between the pooled sample's (2 bees per) abdominal FAS activity and HPG acini area ($n$ = 35 pooled bees). Per treatment, 18 bees were used except for the case of the 3-day-old QMP + treatment group in the first replicate as there was a dissection issue. An additional sample was measured but not included in the regression analysis.

In our experiment, caged 8-day-old worker bees had larger amounts of abdominal protein than 3-day-old bees, while QMP had no effect on this protein level. Protein levels generally increase in young workers after they emerge from pupation [7], suggesting that our caged workers were able to obtain adequate nutrition. More specifically, abdominal protein levels are correlated with the amount of Vitellogenin (Vg) protein in colony-living worker bees [19]. Vg is an important indicator of nutrition and health in honey bees, and the protein influences several aspects of worker physiology and behavior, including the function of the HPGs [19]. We did not measure Vg in our experiment, but we did measure total abdominal protein. HPG size was not associated with measurable changes in food consumption metrics or abdominal FAS activity but was correlated with the abdominal protein content. Consumption of adequate dietary protein is necessary for the development of nurse bee physiology, such as large HPGs [43]. The correlation between abdominal protein and HPG size supports the link between the protein status of the bee and nursing status. This suggests that there is variation between bees in how much protein they consume and thus have available for synthesis of Vg in the fat body which can then be transported to the HPGs to be used for jelly synthesis [19].

Our experiment detected a significantly increased HPG acini size in both 3d and 8d nurse-aged bees exposed to QMP. QMP has been previously shown to suppress levels of circulating juvenile hormone (JH) [4, 45] and treatment with JH analog will reduce HPG size [46, 47]. Vg production is also known to be increased by QMP exposure [12], potentially due to a reduction in the inhibitory signal from JH. Thus, one explanation is that QMP allowed HPGs to develop because JH was suppressed in this treatment group. In addition to this JH effect, there may be other undiscovered mechanisms responsible for the observed pattern in our study.

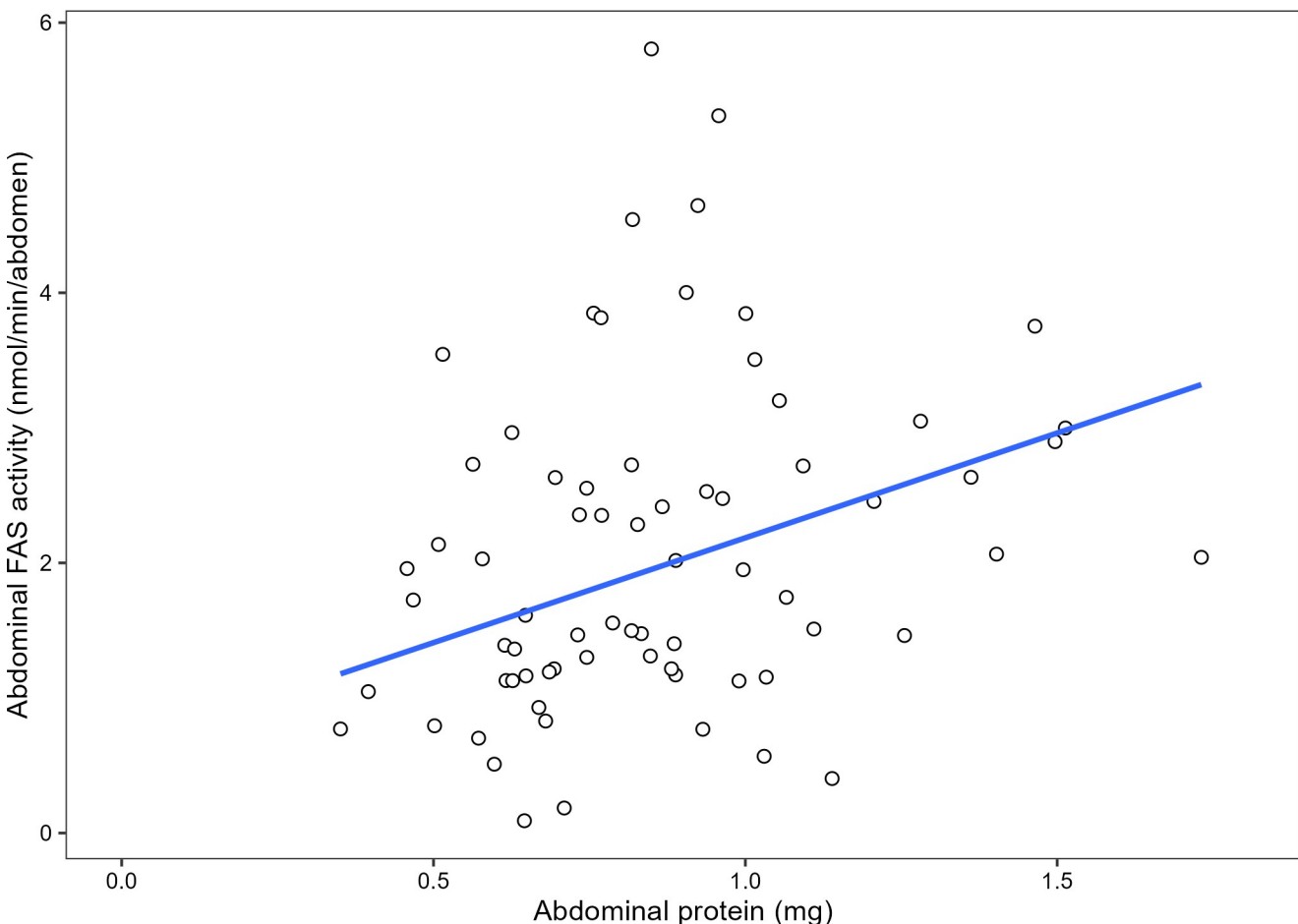

**Fig 11. Non-parametric linear regression scatter plot between abdominal protein (mg) and abdominal FAS (nmol oxidized NADPH/min).** Abdominal FAS activity was significantly predicted by abdominal protein quantity (Kendall-Theil Sen Siegel linear regression: estimate = 1.5049 ± 1.4016, $V$ = 2265, Efron's pseudo $R^2$ = 0.0498, $P$ < 0.001). Each point represents the correlation between the pooled sample's (2 bees per) abdominal protein and abdominal FAS. Per treatment, 36 bees were used, resulting in $n$ = 72 pooled bees and a total of 144 abdomens.

It is known that HPGs contain a high concentration of lipids [15]. This, combined with nurses having high fat body lipids relative to foragers, suggests that HPG lipids are important for jelly production. Our study aimed to determine if lipogenic capacity is increased in nurses' abdominal fat bodies in the presence of QMP, if this increased fat body lipogenesis functions

**Table 1. Regression analysis of $P$ values between all combinations of abdominal protein, abdominal FAS activity and HPG acini area.**

|  | Abdominal protein | Abdominal FAS activity | HPG acini area |
|---|---|---|---|
| Abdominal protein |  | $9.61 \times 10^{-8}$ *** | 0.00116** |
| Abdominal FAS activity | $9.61 \times 10^{-8}$ *** |  | 0.02123* |
| HPG acini area | 0.00116** | 0.02123* |  |

Asterisks represent significant differences (* denotes P < 0.05

** denotes $P$ < 0.01

*** denotes $P$ < 0.001) for each correlation. The sample size taken for each regression analysis was $N$ = 35, except for abdominal protein and abdominal FAS which was $N$ = 72. The full regression analysis statistics can be found in S1 File. Normalized FAS activity was not included in this matrix as it was used as a summary statistic and will co-vary from the factors that it was calculated from (abdominal protein/abdominal FAS activity).

to support HPG development, and whether these two pathways are connected. Using a lipid-deficient diet to eliminate the possibility of QMP increasing nurse-aged bees' consumption of lipids, we found that QMP significantly increased HPG size without affecting nurse-aged lipogenic capacity, suggesting that the fat body lipids levels are not as important for the development of HPGs compared to abdominal protein levels. However, a limitation of our study is that we only assessed HPG size, not content or secretory activity, so it is possible that there is some relationship between fat body FAS activity and HPG function that we did not address here. For example, bees that are fed protein-containing diets develop both large HPGs and high abdominal FAS activity [43]. A recent study showed that treatment with QMP strips causes 8-day-old bees to have higher HPG expression of major royal jelly protein 1, the most abundant protein in royal jelly, supporting the idea that the increased HPG size we found in this study also results in increased jelly production [11]. Overall, this study emphasized the importance of how pheromonal regulation can influence adaptive physiology and nutrient storage in worker honey bees. This knowledge provides further insight into basic bee biology and to queen pheromone, which is commercially used in apiculture.

## Supporting information

**S1 Fig. The relationship between age/QMP treatment and mortality of bees in %.**
(DOCX)

**S2 Fig. The effects of replicates on abdominal FAS activity, abdominal protein, normalized FAS activity, and HPG acini area.**
(DOCX)

**S1 File. The full regression plot analysis statistics between abdominal protein and HPG acini area, abdominal FAS and HPG acini area, and abdominal protein and abdominal FAS.**
(DOCX)

## Acknowledgments

We would like to thank Cahit Ozturk for providing us with the honeybees for this experiment, Amalie Strange for assisting us on how to perform the bee head dissections, Jenna Dobson for blindly choosing the acini, and Matthew Prebus for his training on how to use the Leica microscope. This research was supported by the Barrett Thesis Funding Grant from the Barrett College at Arizona State University. Special thanks to Christine Fleetwood from the Barrett College and Maricel Scalzo from the School of Life Sciences Business Office for assisting with the Thesis funding and reimbursement.

## Author Contributions

**Conceptualization:** Angela Oreshkova, Sebastian Scofield, Gro V. Amdam.

**Data curation:** Angela Oreshkova, Sebastian Scofield.

**Formal analysis:** Angela Oreshkova, Sebastian Scofield.

**Funding acquisition:** Angela Oreshkova.

**Investigation:** Angela Oreshkova, Sebastian Scofield.

**Methodology:** Sebastian Scofield.

**Project administration:** Angela Oreshkova, Gro V. Amdam.

**Resources:** Gro V. Amdam.

**Supervision:** Gro V. Amdam.

**Validation:** Gro V. Amdam.

**Visualization:** Angela Oreshkova.

**Writing – original draft:** Angela Oreshkova.

**Writing – review & editing:** Angela Oreshkova, Sebastian Scofield, Gro V. Amdam.

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
