## [Decision Letter · Decision Letter 0]

30 Oct 2023

PONE-D-23-30810

The effects of queen mandibular pheromone on nurse-aged honey bee (* Apis mellifera *) hypopharyngeal gland size and lipid metabolism

PLOS ONE

Dear Dr. Oreshkova,

Thank you for submitting your manuscript to PLOS ONE. After careful consideration, we have decided that your manuscript does not meet our criteria for publication and must therefore be rejected.

Specifically:

It was extremely difficult to find reviewers willing to review the manuscript - please, make sure to suggest reviewers in a future submission. I also agree with the comments from the reviewer, especially that given the modest sample size and study design flaws can impact the findings and conclusions reported. Although this is an interesting topic, the work itself is lacking quality to be published by PLOS ONE.

I am sorry that we cannot be more positive on this occasion, but hope that you appreciate the reasons for this decision.

Kind regards,

Laura Patterson Rosa, M.V., Ph.D.

Academic Editor

PLOS ONE

Additional Editor Comments (if provided):

Reviewers' comments:

Reviewer's Responses to Questions

**Comments to the Author**

1. Is the manuscript technically sound, and do the data support the conclusions?

Reviewer #1: Partly

2. Has the statistical analysis been performed appropriately and rigorously? 

Reviewer #1: Yes

3. Have the authors made all data underlying the findings in their manuscript fully available?

Reviewer #1: Yes

4. Is the manuscript presented in an intelligible fashion and written in standard English?

Reviewer #1: Yes

5. Review Comments to the Author

Reviewer #1: Main commentary:

Queen mandibular pheromone (QMP) increases lipids in the abdomen and proteins in hypopharyngeal glands (HPGs). The authors hypothesized that QMP increases lipid synthesis and HPG size. The results suggest that QMP does not increase lipid synthesis but increases HPG size in a free-lipid protein diet. While the positive effect of QMP on HPG size was previously reported, the finding that such effect does not depend on lipid ingestion is novel. However, the study's experimental design has limitations that reduce the scope of the findings.

General comments:

1) While the authors controlled important experimental variables, several important factors are missing in this experiment. One of them is the absence of trophallactic intersections between the caged bees and older bees in the colony. This lack of trophallactic interactions results in altered microbiota and reduced transfer of food (e.g., MRJPs), which significantly affect the nutritional conditions of the bees under study and potentially the results obtained.

2) Studies analyzing complex social interactions among honey bee castes could require a sufficient number of bees to have a better chance of capturing a realistic result. Thirty bees per cage and three replicates per treatment is a modest experimental size for this type of study.

3) Additional controls such as other diets (e.g., pollen) could have benefited the study.

4) What was the mortality rate of the bees under the different treatments? Was this significant?

Specific points:

Methods.

Ln 105. Correct the dates when the experiments were performed.

Ln 164. How many bees per cage/treatment were analyzed?

6. PLOS authors have the option to publish the peer review history of their article (what does this mean?). If published, this will include your full peer review and any attached files.

Reviewer #1: No

- - - - -

---

## [Author Response · Author response to Decision Letter 0]

17 Dec 2023

Dear reviewer/editor,

Thank you for the feedback on our submission to PLOS ONE and thank you for the opportunity to submit

a revised version of our manuscript. We were pleased to receive thoughtful comments that are easy to

address. Down below we have responded to each of your concerns with details on the revisions carried

out on our manuscript since its original submission.

1. Reviewer comment: “While the authors controlled important experimental variables, several

important factors are missing in this experiment … [these] significantly affect the nutritional

conditions of the bees under study and potentially the results obtained.” 

Authors’ reply: Cage experiments that eliminate these factors are standard practice in the field,

specifically when investigating the role of pheromones and diet on physiological parameters, please

see examples: [1]–[3]. We will provide this information in a new sentence on line 116: “To eliminate

these issues and provide greater insight into specific nutrients in pollen, a number of recent studies

have used artificial diets [16, 23, 24].”

2. Reviewer comment: “Thirty bees per cage and three replicates per treatment is a modest

experimental size for this type of study.”

Authors’ reply: While we realize there is discussion about the appropriate sample size for

consumption metrics, the number of bees we used is standard for inferences on physiological

relationships [3]–[5]. We will explain these facts by inserting a clarifying sentence on line 119: “Cage

experiments using cohorts of approximately 30 bees per cage maintained in an incubator are

standard practice when investigating the role of pheromones and diet on physiological parameters

and have been used in a number of different studies [16, 21, 25].”

3. Reviewer comment: “Additional controls such as other diets (e.g., pollen) could have benefited the

study.”

Authors’ reply: Many articles address roles of dietary pollen, e.g. [6]. Our goal was not to re-do these

studies. Instead, we sought to optimize data reliability: Pollen is a natural compound that is highly

variable and difficult to standardize. Our experiment eliminates this variability, thus allowing for

greater reliability of data. We will clarify this context by inserting a sentence on line 113: “We chose

to use artificial diets because many studies have shown strong effects of dietary pollen on worker

physiology [11], but pollen composition is highly variable and difficult to standardize [21] and pollen

may be contaminated with pesticides [22].”

4. Reviewer comment: “What was the mortality rate of the bees under the different treatments? Was

this significant?”

Authors’ reply: Mortality was less than 7% over the course of the experiment. This is within the

accepted range for honey bee cage studies [7]. This information was added on line 212: “Mortality

rates were low (less than 7%) and within accepted ranges for cage experiments [19] for all ages and

did not differ by QMP treatment (Kolmgorov-Smirnov, D(10) = 0.167, P = 1; Fig S1).” A new

supplemental figure was added to the manuscript as well.

Thank you for reviewing our revisions. We would be pleased to assist you with securing additional

reviewers for the manuscript if needed,

On behalf of the author team:

Angela Oreshkova (12/14/2023)

[1] D. Stabler, M. Al-Esawy, J. A. Chennells, G. Perri, A. Robinson, and G. A. Wright, “Regulation of dietary intake of protein and lipid by nurse-age

adult worker honeybees,” J. Exp. Biol., p. jeb.230615, Jan. 2020, doi: 10.1242/jeb.230615.

[2] A. M. Trawinski and S. E. Fahrbach, “Queen mandibular pheromone modulates hemolymph ecdysteroid titers in adult Apis mellifera workers,”

Apidologie, vol. 49, no. 3, pp. 346–358, Jun. 2018, doi: 10.1007/s13592-018-0562-6.

[3] P. Fischer and C. M. Grozinger, “Pheromonal regulation of starvation resistance in honey bee workers (Apis mellifera),” Naturwissenschaften, vol. 95, no. 8, pp. 723–729, Aug. 2008, doi: 10.1007/s00114-008-0378-8.

[4] E. Omar, A. A. Abd-Ella, M. M. Khodairy, R. Moosbeckhofer, K. Crailsheim, and R. Brodschneider, “Influence of different pollen diets on the

development of hypopharyngeal glands and size of acid gland sacs in caged honey bees (Apis mellifera),” Apidologie, vol. 48, no. 4, pp.

425–436, Jul. 2017, doi: 10.1007/s13592-016-0487-x.

[5] C. M. Grozinger and G. E. Robinson, “Endocrine modulation of a pheromone-responsive gene in the honey bee brain,” J. Comp. Physiol. A

Neuroethol. Sens. Neural. Behav. Physiol., vol. 193, no. 4, pp. 461–470, Mar. 2007, doi: 10.1007/s00359-006-0202-x.

[6] V. Corby-Harris et al., “Diet and pheromones interact to shape honey bee (Apis mellifera) worker physiology,” J. Insect Physiol., vol. 143, no.

October, p. 104442, 2022, doi: 10.1016/j.jinsphys.2022.104442.

[7] S. K. Huang et al., “Evaluation of cage designs and feeding regimes for honey bee (Hymenoptera: Apidae) laboratory experiments,” J. Econ.

Entomol., vol. 107, no. 1, pp. 54–62, Feb. 2014, doi: 10.1603/EC13213.

---

## [Decision Letter · Decision Letter 1]

3 Apr 2024

PONE-D-23-30810R1The effects of queen mandibular pheromone on nurse-aged honey bee (* Apis mellifera *) hypopharyngeal gland size and lipid metabolismPLOS ONE

Dear Dr. Oreshkova,

Thank you for resubmitting your manuscript to PLOS ONE and I apologize for the performance of the previous editor. After careful consideration and two additional reviews, we feel that your manuscript has merit but does not fully meet PLOS ONE’s publication criteria as it currently stands. Therefore, we invite you to submit a revised version of the manuscript that addresses the points raised during the review process. Given the small sample size, a more modest interpretation and deliberate discussion of limitations is warranted. There are numerous specific comments that also need to be addressed. 

We look forward to receiving your revised manuscript.

Kind regards,

Olav Rueppell

Academic Editor

PLOS ONE

Reviewers' comments:

Reviewer's Responses to Questions

**Comments to the Author**

1. If the authors have adequately addressed your comments raised in a previous round of review and you feel that this manuscript is now acceptable for publication, you may indicate that here to bypass the “Comments to the Author” section, enter your conflict of interest statement in the “Confidential to Editor” section, and submit your "Accept" recommendation.

Reviewer #2: (No Response)

Reviewer #3: (No Response)

2. Is the manuscript technically sound, and do the data support the conclusions?

Reviewer #2: Partly

Reviewer #3: Partly

3. Has the statistical analysis been performed appropriately and rigorously? 

Reviewer #2: Yes

Reviewer #3: Yes

4. Have the authors made all data underlying the findings in their manuscript fully available?

Reviewer #2: (No Response)

Reviewer #3: Yes

5. Is the manuscript presented in an intelligible fashion and written in standard English?

Reviewer #2: Yes

Reviewer #3: Yes

6. Review Comments to the Author

Reviewer #2: This manuscript presents the findings of a small study examining the effects of QMP on lipogenic activity in the fat bodies of bees and on HPG development. The findings are interesting and generally well explored in the discussion section, though I do agree with a previous reviewer that there are some limitations to the interpretation due to experimental design. In my opinion, these limitations do not invalidate the findings, however they do warrant some discussion. I think the sample size used here was rather small. The authors do a good job discussing the implications of their small sample size on the results of their diet consumption analyses, but might this have implications for other aspects of their study? If the authors don’t think so, please add some explanation as to why this was unlikely to have been an issue to the discussion. I also agree that using only one diet is a limitation. It would have been interesting to see how FAS activity differed depending on nutrient availability, and I think some discussion on the subject would make for a more satisfying discussion. Finally, the authors looked at HPG size but not content. Could protein/lipid composition vary in a way that is related to FAS activity? I have a few other specific comments. See below for details:

L36: A couple of points that may seem unimportant but are somewhat fundamental: HPGs secrete worker jelly and royal jelly, which are different. This should be adjusted in your abstract and when it is mentioned again in the text.

Wang, Y., Ma, L., Zhang, W., Cui, X., Wang, H., Xu, B., 2016. Comparison of the nutrient composition of royal jelly and worker jelly of honey bees (Apis mellifera). Apidologie 47, 48–56. https://doi.org/10.1007/s13592-015-0374-x

Likewise, large HPGs do not always signify nurse bee-like physiology. Winter bees also have enlarged HPGs.

L134: What dose (Qeq’s) does this synthetic lure represent? This can make a big difference in your experiment. The authors do acknowledge that the use of synthetic QMP may have affected their results, but could dose have played a role?

L171: Starting sentences with a number should be avoided.

L191: When Kruskal-Wallis tests were used, did you only compare QMP treatment within age groups and age groups within QMP treatment?

L227: It might not be relevant to look at FAS activity without first normalizing it to abdominal protein content. What is the standard practice for this measurement? Consider eliminating the unnormalized data from this ms.

L333: I might not be fully understanding this statement, but I’m not sure that the authors fully examined whether workers increase HPG size through increased food consumption. They only used a suboptimal, low lipid diet. It’s possible that consumption rates and FAS activity might differ depending on diet composition and QMP presence.

L360: I don’t really follow this logic. It seems far more likely that there is some interaction between diet and QMP that isn’t captured by this experiment.

L378: I’m not really seeing the connection between this statement and reference 20.

Reviewer #3: All my comments are in the attached document. Please see "reviewer comments to authors" for my feedack.

7. PLOS authors have the option to publish the peer review history of their article (what does this mean?). If published, this will include your full peer review and any attached files.

Reviewer #2: No

Reviewer #3: No

---

## [Author Response · Author response to Decision Letter 1]

1 May 2024

Dear reviewers/editor,

Thank you for the feedback on our submission to PLOS ONE and thank you for the opportunity to submit a revised version of our manuscript. We were pleased to receive thoughtful comments that we are able to address. Down below we have responded to each of your concerns with details on the revisions carried out on our manuscript since its original submission (the line numbers refer to the track changes manuscript document):

1. Reviewer #1 comment: “… The authors do a good job discussing the implications of their small sample size on the results of their diet consumption analyses, but might this have implications for other aspects of their study? If the authors don’t think so, please add some explanation as to why this was unlikely to have been an issue to the discussion. … It would have been interesting to see how FAS activity differed depending on nutrient availability… Finally, the authors looked at HPG size but not content. Could protein/lipid composition vary in a way that is related to FAS activity?”

Authors’ reply: We understand that our original submission was unclear in its communication of sample sizes. In contrast to the small sample size for the consumption results, we had a large sample size for the other components of the study, for example 72 total samples for the FAS activity. We added discussion of sample size to L428. We agree that the effects of dietary nutrition on FAS activity are worth exploring though outside the scope of this study. We have addressed this in a separate study and added a statement and citation explaining this on L422. We added a statement addressing the limitation of our study in not measuring FAS content or secretory activity on L485.

2. Reviewer #1 comment: “HPGs secrete worker jelly and royal jelly, which are different. This should be adjusted in your abstract and when it is mentioned again in the text. Likewise, large HPGs do not always signify nurse bee-like physiology. Winter bees also have enlarged HPGs.”

Authors’ reply: It is true that the composition of worker and royal jelly are different, as noted in the reference provided by the reviewer. To avoid confusion, we replaced “royal jelly” by the term “proteinaceous jelly” in lines 35, 36, and 79 as both types of jelly contain protein as the primary macronutrient [1]. To further take action on this important comment, we added a new sentence on line 474: “It is known that HPGs contain a high concentration of lipids [15]. This, combined with nurses having high fat body lipids relative to foragers, suggests that HPG lipids are important for jelly production.”

3. Reviewer #1 comment: “L134: What dose (Qeq’s) does this synthetic lure represent?...The authors do acknowledge that the use of synthetic QMP may have affected their results, but could dose have played a role?”

Authors’ reply: We understand that our original submission was unclear in what dosage of synthetic lure we used and how the dosage could have played a role in our study. Thus, we added details on Qeqs to L157 and justification of the application and dose to L158.

4. Reviewer #1 comment: “L175: Starting sentences with a number should be avoided.”

Authors’ reply: We agree that starting sentences with a number should be avoided, thus, we corrected line 195 to now read: “Per treatment group, 18 heads were dissected, resulting in a total of 72 heads.”

5. Reviewer #1 comment: “When Kruskal-Wallis tests were used, did you only compare QMP treatment within age groups and age groups within QMP treatment?”

Authors’ reply: We used the Kruskal-Wallis test to compare all four treatment groups (combination of QMP and age) rather than comparing QMP and age separately. This was clarified on line 216.

6. Reviewer #1 comment: “L227 … Consider eliminating the unnormalized data from this ms.”

Authors’ reply: While we agree that eliminating redundant figures is important, there are two important reasons to keep the unnormalized data: first is that the normalization factor itself is significantly different between age groups, potentially confounding interpretation. For full transparency, we want to show the unnormalized trend here. Second is that the unnormalized data are themselves meaningful, since we are reporting the total lipogenic capacity in two bee abdomens. Reporting these data (such as total lipids per abdomen) are standard practice in honey bee research and useful for aiding in interpretation of the data.

7. Reviewer #1 comment: “I’m not sure that the authors fully examined whether workers increase HPG size through increased food consumption. They only used a suboptimal, low lipid diet. It’s possible that consumption rates and FAS activity might differ depending on diet composition and QMP presence.”

Authors’ reply: We agree that consumption rates differ depending on diet composition and QMP presence as highlighted in two studies mentioned in our manuscript [2, 3]. We further explained the limitations of our experiment by inserting the following statements on line 420: “Furthermore, our study used a lipid deficient diet while Ament and colleagues [13] and Peters and colleagues [14] used a combination of diets consisting of only pollen, pollen and sucrose, or only sucrose. There may be some interacting effects between QMP and nutrition which we did not test for.”

8. Reviewer #1 comment: “It seems far more likely that there is some interaction between diet and QMP that isn’t captured by this experiment.”

Authors’ reply: We agree that there are limitations to our experiment as we only studied one type of diet. We inserted a statement on line 441 to explain how diet (among other factors) may contribute to the differing results of QMP affecting abdominal lipid stores: “If the activity levels and lipid catabolism rates are different between our bees and the bees used in the other studies [11,12,13] due to varying ages, environment, and diet, this could explain the inconsistency of whether QMP affects abdominal lipid stores.”

9. Reviewer #1 comment: “I’m not really seeing the connection between this statement and reference 20.”

Authors’ reply: We apologize that the incorrect reference was cited in the original submission. The correct reference to the statement on line 472 is reference 44. However, we have decided to remove this statement from the revised manuscript as we felt it did not provide relevant information to the passage. 

10. Reviewer #2 comment: “Please include whether experimental units are cages or bees and provide sample sizes in the summary statistics for how many bees of each treatment were used. This experiment has an n=3 cages per treatment. This is fairly low but could perhaps be justified with clear explanation in the methods … The authors should make this clear and provide proper explanation and justification. Yet, some of the results described in this paper are cage-level results.”

Authors’ reply: We understand that our original submission was unclear of what experimental units we used. We rewrote the passage on line 228 to clarify our experimental units and provide justification to our low consumption metric sample size: “For all measurements except for consumption, the experimental unit was bees or pooled bees. For consumption metrics, the cage was the experimental unit. A balanced study design was employed for all factors. The sample size for cage level consumption metrics was low at n = 3 cages per treatment as this was not the focus of our study and the metrics describe the physiological parameters of the bees.”

11. Reviewer #2 comment: “These results (food consumption, mortality) are not the central findings of the paper, but they are still discussed … However, a lack of significance could come from the very low sample size, and is not necessarily evidence that these metrics would not differ between treatments if the sample sizes were larger. This should be mentioned in the discussion/interpretation.”

Authors’ reply: To clarify that a lack of significance between protein/sucrose consumption among the four treatment groups may be due to our low sample size, we revised line 417 to now read: “The lack of significant differences in food consumption between our treatments could be due to our small sample size. Thus, Because our study had the smallest sample size, our data should be interpreted cautiously.”

12. Reviewer #2 comment: “A more appropriate chart type should be used instead [for food consumption data].”

Authors’ reply: We agree that a more appropriate chart type should be used as the sample size for food consumption data was below 5. Thus, we revised figures 3 and 4 to now be dot plots rather than box plots. Please refer to the new attached figures for these changes. 

13. Reviewer #2 comment: “Why does the ‘natural food composition make it difficult to tease apart how QMP acts to influence physiology’? It’s not clear what is meant by “natural food composition” or how it relates to QMP.”

Authors’ reply: We understand that we were not clear what ‘natural food consumption’ means. We define natural food consumption as the “variable macronutrient composition (lipids and amino acids) of pollen as well as the potential contamination of pesticides” and this was added to line 83. Furthermore, we clarified how consumption relates to QMP with a reference added to line 81: “Furthermore, there has been research that exposure to QMP increases both pollen and sucrose solution consumption in nurse-aged bees [13].” 

14. Reviewer #2 comment: “Line 85: Are there many possibilities? I suggest either outlining some more potential possibilities, or removing this final line of the paragraph.”

Authors’ reply: We understand that ending this passage openly is vague and can confuse the reader of whether there are more possibilities outside of what we mentioned. Therefore, the final sentence of line 89 was removed for concision and we ended the paragraph on line 87 with our question of: “Are the QMP-induced increases in lipid stores and HPG size behaviorally modulated by increasing consumption of lipid-containing pollen, or are they due to effects on metabolic pathways regulating lipid and protein synthesis and storage?”

15. Reviewer #2 comment: “Can you please define lipogenic capacity here?

Authors’ reply: We understand that we did not define lipogenic capacity well in our original submission and thus added a brief definition of lipogenic capacity to L96.

16. Reviewer #2 comment: “Why does your hypothesis make predictions about HPG size? Can you better explain your reasoning for why you hypothesize that increased lipogenic capacity would mean larger HGPs?

Authors’ reply: We understand that our hypothesis reasoning for why increased lipogenic capacity would mean larger HPGs could be explained better. We added additional explanations and citations to L99-105.

17. Reviewer #2 comment: “Why did you use these two different ages? Can you outline your hypothesis/predictions for 3-day vs. 8-day old bees?”

Authors’ reply: We added an explanation of our reasoning for using two age groups on L115.

18. Reviewer #2 comment: “(Fig 1 legend): What cages are these dimensions for? Are they different from the Plexiglas cages described in line 109 of the methods?”

Authors’ reply: We apologize for the confusion in cage dimensions. We remeasured and inputted the correct cage dimensions on line 128 and 145 which now read: “The dimensions measured were 16 cm × 12 cm × 9 cm. Because approximately half the cage was used, the effective depth was 6 cm.”

19. Reviewer #2 comment: “These sample sizes (3 cages per treatment) seem fairly low. Although the authors addressed a previous reviewer’s concerns about the number of bees per cage (line 120), can you also include justification for the small replicate size?”

Authors’ reply: We understand that our original submission did not discuss the justification for our replicate size. We added justification of the level of replication to L150.

20. Reviewer #2 comment: “The sample sizes were n=3. Does this mean 3 per treatment, or 3 total? What is the makeup of these sample sizes (18, 35, and 72)? Are they pooled within cages or across cages? How are cages and treatments represented within these sample sizes? Are they balanced? If not, please give specifics of how many bees were used from each treatment, per cage.”

Authors’ reply: We understand that the sample size section in our original submission was not adequately explained. We rewrote the passage beginning on line 228 to clarify the makeup of our sample sizes and whether they are balanced: “For all measurements except for consumption, the experimental unit was bees or pooled bees. For consumption metrics, the cage was the experimental unit. A balanced study design was employed for all factors. The sample size for cage level consumption metrics was low at n = 3 cages per treatment as this was not the focus of our study and the metrics describe the physiological parameters of the bees. For measuring abdominal FAS activity, normalized FAS activity, abdominal protein, and HPG acini area, 2 bees were pooled for each biological sample and 6 samples totaling 12 bees were collected from each cage for all four treatment conditions. This totaled 72 biological samples pooled from 144 abdomens as three replicates were performed. Thus, the sample size was n = 18 pooled samples per treatment group for these three parameters. The sample size for the regression analyses was n = 35 pooled samples as there was a dissection issue for one of the bees and only half of the HPGs of the 24 pooled samples per replicate were measured, except for the case of abdominal protein and abdominal FAS which was n = 72 pooled samples (144 total abdomens).”

21. Reviewer #2 comment: “Line 206-214: Please include sample sizes for test statistics. Also, please explain ‘between replicates’: Are these stats explanatory for all 4 treatments?” “Line 233: ‘significant differences between the three replicates’: For which treatment? This needs to be clear each time a significant difference is reported between replicates (like in line 237, for example).”

Authors’ reply: We included sample sizes for food and sucrose consumption on lines 245 and 247. We apologize for not outlining our replicate effects clearly in the original manuscript. To address the reviewers' comments to lines 206 - 214 and line 233, we have included a detailed explanation of what the replicate effect is beginning on line 219: “For each parameter analyzed with either a parametric or non-parametric test, replicate effects were tested for. These describe whether there were significant differences between each replicate and if there was an independent effect of doing a replication of the experiment overall.”

22. Reviewer #2 comment: “Figures 3 and 4, and their legends: include ‘(days)’ in x-axis labels; These figure legends … Either a simple point plot or bar charts would be more honest visual representations of the data.”

Authors’ reply: We included ‘days’ in the x-axis labels for figures 3 and 4 (lines 253 and 259). We agree that a more appropriate chart type should be used as the sample size for food consumption data was below 5. Thus, we revised figures 3 and 4 to now be dot plots rather than box plots. Please refer to the new attached figures for these changes. 

23. Reviewer #2 comment: “Were pooled bees from the same cage? Were there 6 pools per cage? Figures 5, 6, 7, and 8 legends: ‘Outliers are points outside the maximum and minimum distribution.’] How do you have points outside the max and min of the distribution? Do these fall outside the expected variation of the data?”

Authors’ reply: We agree that sample sizes should better be explained for figure 5,6,7, and 8 legends. We clarified how the sample size was calculated for these figures with the following sentence added to lines 285, 298, 321, and 343: “As there were 6 pools per cage (12 abdomens/bees sampled) and three replicates performed, the sample size is n = 18 pooled samples.” We amended the figure legends to explain that whiskers are calculated as 1.5 times the interquartile range and outliers are data points outside the whiskers that depict the expected variation of the data.

24. Reviewer #2 comment: “Fig 10/11 legend: Provide sample sizes in the summary

---

## [Decision Letter · Decision Letter 2]

13 May 2024

PONE-D-23-30810R2The effects of queen mandibular pheromone on nurse-aged honey bee (* Apis mellifera *) hypopharyngeal gland size and lipid metabolismPLOS ONE

Dear Dr. Oreshkova,

Thank you for submitting your manuscript to PLOS ONE. Both reviewers find your manuscript much improved and thus, it is almost ready for acceptance. However, I would like to give you an additional opportunity to take the comments of the reviewers into account for improving your manuscript further.

We look forward to receiving your revised manuscript.

Kind regards,

Olav Rueppell

Academic Editor

PLOS ONE

Journal Requirements:

Reviewers' comments:

Reviewer's Responses to Questions

**Comments to the Author**

1. If the authors have adequately addressed your comments raised in a previous round of review and you feel that this manuscript is now acceptable for publication, you may indicate that here to bypass the “Comments to the Author” section, enter your conflict of interest statement in the “Confidential to Editor” section, and submit your "Accept" recommendation.

Reviewer #2: (No Response)

Reviewer #3: All comments have been addressed

2. Is the manuscript technically sound, and do the data support the conclusions?

Reviewer #2: Partly

Reviewer #3: Yes

3. Has the statistical analysis been performed appropriately and rigorously? 

Reviewer #2: I Don't Know

Reviewer #3: Yes

4. Have the authors made all data underlying the findings in their manuscript fully available?

Reviewer #2: Yes

Reviewer #3: Yes

5. Is the manuscript presented in an intelligible fashion and written in standard English?

Reviewer #2: Yes

Reviewer #3: Yes

6. Review Comments to the Author

Reviewer #2: I appreciate that the authors have addressed most of my concerns. This is a much improved manuscript in terms of clarity, however I have several more comments I feel should be addressed before the manuscript is suitable.

Please note, line numbers refer to the tracked changes version of the manuscript revision.

Please represent significant differences somehow in your figures.

Standard practice for analyzing FAS activity is to normalize it to protein content largely because protein content will vary between samples. You may be seeing a relationship between FAS activity and abdominal protein content because FAS is partially dependent on amino acids and carbohydrates as substrates. Why wouldn’t you depict it the way you analyzed it in your figures? And why isn’t it represented this way in your table?

N=18 is a reasonable sample size for FAS analysis but it’s small for HPGs. The authors continually state that 18 is a relatively large sample size. Relative to what?

I pointed out in my last review that it isn’t grammatically correct to begin a sentence with a number. There are several new instances of this error in this version that need to be addressed.

L216-222: I'm still unclear on some aspects of the stats. I see that you used KW or anova depending on whether the data satified assumptions. How were replicate effects evaluated for KW models? Did you just run separate KW tests for each rep? For ANOVA models, was replicate treated as a factor? Did you do anything to account for possible interaction between factors for ANOVA and KW models? I imagine there were limitations associated with using the KW models since you can't run the equivalent of a 2-way anova. How did you try to address those?

L233-238: All of this should perhaps be described earlier, when discussing FAS activity assay methods.

Reviewer #3: Thank you for addressing my comments. One final note, please add a definition of "queen equivalent" (line 157).

7. PLOS authors have the option to publish the peer review history of their article (what does this mean?). If published, this will include your full peer review and any attached files.

Reviewer #2: No

Reviewer #3: No

---

## [Author Response · Author response to Decision Letter 2]

4 Jul 2024

Dear reviewers/editor,

Thank you for the additional feedback on our submission to PLOS ONE and thank you for the opportunity to submit a second revised version of our manuscript. We were pleased to receive additional thoughtful comments that we are able to address. Down below we have responded to each of your new concerns with details on the revisions carried out on our manuscript since its original submission (the line numbers refer to the new track changes manuscript document):

1. Reviewer #2 comment: “Please represent significant differences somehow in your figures.”

Authors’ reply: We agree that significant differences should be represented in our figures. Please refer to attached figures 6, 7, and 8 to view these changes. Furthermore, we adjusted the legends of these figures (L310, L330 and L351) to make note of the symbols we used in the figures and what they signify.

2. Reviewer #2 comment: “Standard practice for analyzing FAS activity is to normalize it to protein content largely because protein content will vary between samples. You may be seeing a relationship between FAS activity and abdominal protein content because FAS is partially dependent on amino acids and carbohydrates as substrates. Why wouldn’t you depict it the way you analyzed it in your figures? And why isn’t it represented this way in your table?”

Authors’ reply: We agree that standard practice for analyzing FAS activity is to normalize it to protein content. However, we did choose to report both normalized and non-normalized data and have added a clarification to L188 to explain why: “For a more complete picture of the metabolic state of the bee, we report total abdominal FAS activity as well as FAS activity normalized to mg of abdominal protein. Because our dissection approach collects the entire abdominal cuticle and attached fat body, it produces a metric of total lipogenic capacity per abdomen, similar to past studies that have measured total lipids per abdomen using the same dissection technique [7, 12, 15, 16]. To control for potential differences in protein extraction efficiency, we also measure protein concentration in each sample and normalize FAS activity relative to extracted protein. Comparing results from the two metrics makes it clearer when treatment group differences are due to changes in the quantity of activated FAS or due to changes in the normalization factor itself.” We apologize if we were not clear as to why we reported both data sets.

3. Reviewer #2 comment: “N=18 is a reasonable sample size for FAS analysis but it’s small for HPGs. The authors continually state that 18 is a relatively large sample size. Relative to what?”

Authors’ reply: We have removed the reference to large sample size on L398 and altered the wording on L425-426 to make it clear that our HPG sample size is large relative to our cage-level consumption data.

4. Reviewer #2 comment: “I pointed out in my last review that it isn’t grammatically correct to begin a sentence with a number. There are several new instances of this error in this version that need to be addressed.”

Authors’ reply: We apologize that there were several more instances of beginning our sentences with a number. We corrected this on the following lines: L118, L141, L223, L364, L372, and L379.

5. Reviewer #2 comment: “L216-222: I'm still unclear on some aspects of the stats. I see that you used KW or anova depending on whether the data satisfied assumptions. How were replicate effects evaluated for KW models? Did you just run separate KW tests for each rep? For ANOVA models, was replicate treated as a factor? Did you do anything to account for possible interaction between factors for ANOVA and KW models? I imagine there were limitations associated with using the KW models since you can't run the equivalent of a 2-way anova. How did you try to address those?”

Authors’ reply: We added information on L215 and L227 about how we addressed replicate effects and interactions between factors for ANOVA. We added two sentences on L229-L232 outlining our strategy for using a Kruskal-Wallis effect to test for replicate and treatment effects in the normalized FAS data. 

6. Reviewer #2 comment: “L233-238: All of this should perhaps be described earlier, when discussing FAS activity assay methods.”

Authors’ reply: We agree that there should be a description of how 144 bee abdomens were sampled across three replicates in our FAS activity assay methods and apologize if there was any confusion surrounding this. We added a section on L182 to better describe our sampling methods: “This assay was replicated 3 times. In each replicate, 2 bees were pooled for each biological sample and 6 samples totaling 12 bees were collected from each cage for all four treatment conditions. This totaled with a total of 144 abdomens sampled across three replicates.”

7. Reviewer #3 comment: ‘Thank you for addressing my comments. One final note, please add a definition of "queen equivalent" (line 157).’

Authors’ reply: We agree that the term ‘queen equivalent’ should be defined and therefore, we clarified this on L156: “One queen equivalent represents the amount of QMP a mated queen will produce in a 24-hour period and contains 200 mg of ODA, 80 mg of 9-HAD, 20 mg of HOB, and 2 mg of HVA [34]”.

---

## [Editor Report · Decision Letter 3]

9 Jul 2024

The effects of queen mandibular pheromone on nurse-aged honey bee (* Apis mellifera *) hypopharyngeal gland size and lipid metabolism

PONE-D-23-30810R3

Dear Dr. Oreshkova,

We’re pleased to inform you that your manuscript has been judged scientifically suitable for publication and will be formally accepted for publication once it meets all outstanding technical requirements.

Kind regards,

Olav Rueppell

Academic Editor

PLOS ONE
---

## [Editor Report · Acceptance letter]

17 Jul 2024

PONE-D-23-30810R3 

PLOS ONE

Dear Dr. Oreshkova, 

I'm pleased to inform you that your manuscript has been deemed suitable for publication in PLOS ONE. Congratulations! Your manuscript is now being handed over to our production team.

Kind regards, 

on behalf of

Dr. Olav Rueppell 

Academic Editor

PLOS ONE